# Development and Verification of the Effectiveness of a Fine Dust Reduction Planting Model for Socially Vulnerable Area

YunEui Choi [1] , Eunhye Ji [2] and Jinhyung Chon [3,*]

1   OJEong Resilience Institute (OJERI), Korea University, Seoul 02841, Korea; choiuni313@korea.ac.kr
2   Department of Environmental Science & Ecological Engineering, Korea University, Seoul 02841, Korea; jises0325@korea.ac.kr
3   Division of Environmental Science & Ecological Engineering, Korea University, Seoul 02841, Korea
*   Correspondence: jchon@korea.ac.kr

**Abstract:** Creating a green infrastructure that is effective for reducing fine dust is a significant challenge for urban landscape planners. In this study, a fine dust reduction planting model that can be applied to socially vulnerable area was developed, and its effects were verified. Using $PM_{10}$, $PM_{2.5}$, temperature, relative humidity, wind direction, and wind speed measured for approximately one year, the changes in the concentration of fine dust according to the weather conditions were investigated. As a result of the analysis, there was a significant difference in the concentration of fine dust inside and outside the planting zone ($p < 0.05$). In addition, there is a significant difference between the fine dust reduction effect of the multilayered planting model and the single planting model ($p < 0.05$). The paper's main findings are as follows: (1) When the green cover rate is over 50%, the concentration of fine dust is lower than that outside the planting zones. (2) Multilayered planting zones are more effective in reducing the concentration of fine dust than single-structured planting zones. (3) Multilayered planting zones reduce the concentration of fine dust by changing the microclimate. The results of this study can be used as basic data for small urban planting design to reduce fine dust for children's health in socially vulnerable areas.

**Keywords:** particulate matter; single planting; multilayered planting; green infrastructure; small urban planting design

## 1. Introduction

Since the International Agency for Research on Cancer (IARC) under the World Health Organization (WHO) designated fine dust as a first-class carcinogen in 2013 [1], people have become very interested in particulate matter (PM). Long-term exposure to $PM_{2.5}$ (fine particles, with diameters that are generally 2.5 micrometers and smaller) increases the risk of ischemic heart disease and stroke-related death [2]. Children or seniors are more sensitive to exposure to fine PM because they breathe more air per unit weight than regular adults [3]. On days when the level of PM is high, policy limits outdoor activities, which also has a negative impact on outdoor-recreation-related industries [4]. As such, PM not only adversely affects physical and mental health but also negatively affects society and the economy. The average annual $PM_{2.5}$ concentration in Korea is 24.8 $\mu g/m^3$, the highest in the world and the highest among the 36 OECD countries [5]. Among OECD countries, Korea's $PM_{2.5}$ was the same at 24.8 $\mu g/m^3$ in both 2018 and 2019, but Chile, which topped the list in 2018, saw a reduction of $PM_{2.5}$ to 22.6 $\mu g/m^3$ in 2019. To no longer be among the top countries for fine dust levels, Korea has enacted the Special Act on the Reduction and Management of Fine Dust and has implemented long- and short-term fine dust reduction policies for development, industry, road and transportation, and living sectors. As a short-term policy, a fine dust seasonal management system including a reduction in emissions, protection of public health, and cooperation between Korea and China have been implemented. Since the number of high concentration days (over 51 $\mu g/m^3$) and bad

days (above 36 $\mu g/m^3$) of $PM_{2.5}$ account for approximately 72% of days from December to March, the Korean government has set stronger reduction targets than usual and is implementing various regulations during this season. Long-term policies include urban forest creation and management projects to reduce and quickly disperse fine dust in the living sector. The Korea Forest Service has announced 322 types of plants with fine dust reducing effects and encourages their use in urban forest development [6]. Seoul, the capital of Korea, expects fine dust to be reduced through projects to plant 30 million trees by 2022 [7]. Urban forest projects creating green infrastructure such as school forests, roadside trees, and wall greening to reduce fine dust are based on the dust adsorption and blocking functions of plants [8,9]. In Korea, as well as the United States, the United Kingdom, and China, there is a high interest in and expectations for ecofriendly fine dust reduction measures using green infrastructure. Accordingly, a number of studies have been conducted to investigate the fine dust reduction functions of green infrastructure [10–12]. Specifically, previous studies have identified the fine dust accumulation functions of evergreen trees [13], fine dust reduction functions according to tree crown diameter [14], and fine dust absorption functions by the leaf area index [15]. In addition, the effect of fine dust reduction in urban parks planted with trees and shrubs [16,17] and the conversion of the economic value of trees in cities into medical expenses due to fine dust were also studied [18]. However, not all studies on the effect of reducing fine dust with green infrastructure have been positive. Studies have been published that identified negative outcomes on the effect of reducing fine dust with trees depending on the space. Some studies showed that street trees do not reduce the fine dust of a narrow pedestrian path [18,19]. These studies argue that there was not enough space between the street trees and buildings, which blocked air circulation, resulting in an increase in fine dust concentrations on a pedestrian path. In this regard, studies on the relationship between fine dust concentration and wind were conducted, such as the study on the dynamics of air flow by the arrangement of roadside trees [20] and a study of the direction of the creation of wind forests in the city [21]. Moreover, studies on the relationship between urban trees and fine dust concentrations have been conducted from a microscopic perspective focused on only one tree [22–24] or a macroscopic perspective centered on urban structures [25,26]. In the city, trees are planted in various densities and structures according to the characteristics of the space. In the case of roadside trees, if the width of the planting zone is narrow, tall trees are planted in a row at intervals of 8 m, and if the width of the planting zone is wide, tall trees and shrubs are planted in multilayered structures. Because the circulation of air varies depending on the density and structure of the vegetation, fine dust may pass through, be blocked, or be trapped in the planting space. However, until now, there has been insufficient discussion on the effect of reducing fine dust according to the density and structure of trees. Recently, with various activities being restricted by the Coronavirus disease (COVID-19) pandemic, the demand for public open spaces in the living zone has increased. However, old residential areas where low-income families live are densely concentrated and have limited access to park and green spaces [27]. In particular, children are required to participate in a lot of outdoor activities, but they are also vulnerable to fine dust. School forests with fine dust reduction functions can replace the role of parks in socially vulnerable areas. Therefore, we decided that in socially vulnerable urban areas, small green spaces such as school forests can fulfill this task. This study aims to analyze the fine dust reducing effect according to the density and structure of the plants (trees, shrubs, groundcovers, and herbaceous flowers) by proposing a fine dust reduction planting model that can be applied to small urban green infrastructure. This study conducted an empirical experiment in elementary school green areas in an old town where children sensitive to fine dust and reside are active. The results of this study can be used as basic data for planting design to reduce fine dust for children's health in socially vulnerable urban areas.

## 2. Materials and Methods

### 2.1. Study Site

To select vulnerable areas where the fine dust reduction planting model would be applied as a study site, we searched for aged places where residential areas were established 50 years ago, where many low-income families lived, and where there were no children's parks within a radius of 250 m. Since the outdoor space of the school can be both a play space and a rest space for children in an area where there is no children's park, the school was considered first. Dongmyeong Elementary School is a public elementary school opened in 1944 in Seongdong-gu, Seoul, Korea. A total of 345 students between the ages of 6 and 12 attend this school. Most of the area around the school is an aging residential area, with some management areas. About 300 m from the school, there is a small commercial area and a traditional livestock market. The surrounding 500 m radius includes the Cheonggyecheon Stream and an old commercial area (Figure 1a). The area of the school is about 2300 m$^2$, and it consists of the main building, annex, annexed kindergarten, gymnasium, school garden, playground, parking lot, and Seoul public building (Figure 1b). The control and the experimental sites included the planting zone adjacent to the gymnasium.

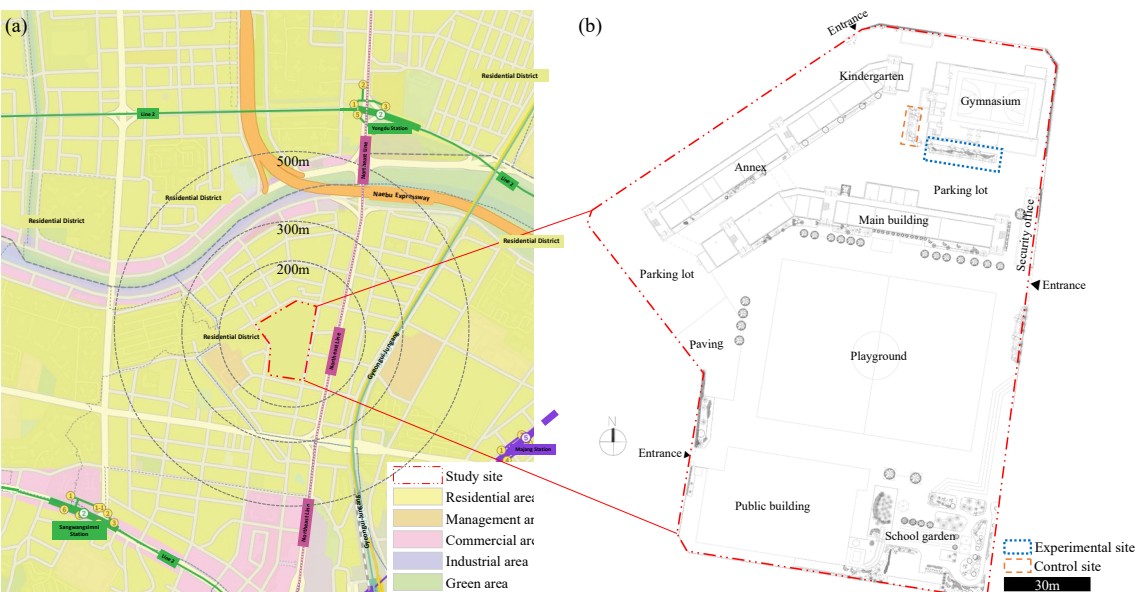

**Figure 1.** The study site: (**a**) Dongmyeong Elementary School is surrounded by a living circle; (**b**) the control and experimental sites are located between the main building and the gymnasium.

### 2.2. Control Site and Experimental Site

The control site was located on the west side of the gymnasium, 12 m in width and 3 m in length, and was approximately 36 m$^2$ in area (Figure 2). The two types of trees in the control site were planted before the experiment in a single structure (Table 1). The bottom of the canopy of the single-structured planting has smooth air flow. To compare the differences between inside and outside of the planting zone, approximately 2 m monitoring poles (No. 1 and No. 2) were installed.

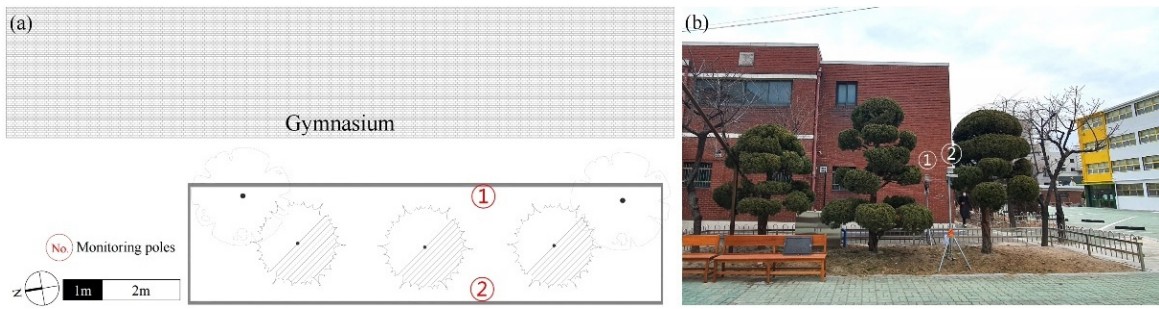

**Figure 2.** Diagram of the control site (**a**) and photo of the control site (**b**).

**Table 1.** Information of plants in control site.

| Division | Symbol | Scientific Name | Size (H:m, R:cm) | No. |
|---|---|---|---|---|
| Evergreen trees | | *Juniperus chinensis* 'Kaizuka' | H3.5 × R20 | 3 |
| Deciduous trees | | *Prunus yedoensis* Matsum. | H4.0 × R20 | 3 |

The experimental site was located on the southern side of the gymnasium, 23.6 m in width and 3.5 m in length, and was approximately 82.6 m² in area (Figure 3). The two types of trees in the experimental site were planted before the experiment in a single structure (Table 2). To compare differences between inside and outside of the planting zone, approximately 2 m monitoring poles (No. 3 and No. 4) were installed. In the parking lot in front of the experimental site, automobile exhaust flows into the gymnasium through the bottom of the canopy. Therefore, we proposed a fine dust reduction planting model that blocks the inflow of exhaust gas from the parking lot. To apply the new planting model, the existing trees of the experimental site were transplanted into the school garden on 13 May 2020.

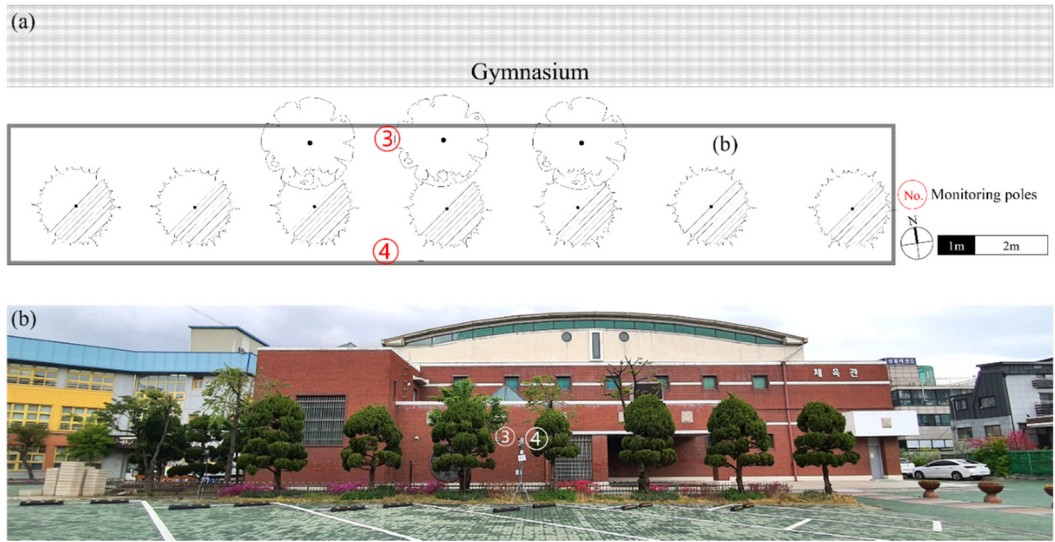

**Figure 3.** Diagram of the experimental site before applying the planting model to reduce fine dust (**a**). Photo of the experimental site before applying the fine dust reduction planting model (**b**).

**Table 2.** Information on plants at the experimental site before applying the fine dust reduction planting model.

| Division | Symbol | Scientific Name | Size (H:m, R:cm) | No. |
|---|---|---|---|---|
| Evergreen trees | | *Juniperus chinensis* 'Kaizuka' | H3.5 × R20 | 7 |
| Deciduous trees | | *Prunus yedoensis* Matsum. | H4.0 × R20 | 3 |

The fine dust reduction planting model was composed of a multilayered structure of evergreen trees, deciduous trees, evergreen shrubs, deciduous shrubs, groundcover plants, and herbaceous flowers that block and absorb fine dust (Figure 4 and Table 3). Trees with fine dust reduction functions are mainly characterized by having hairy leaves, a wax layer, saw-toothed leaf margins, and dense leaves [28].

*Zelkova serrata*, a deciduous broad-leaved tree, has a sawtooth-shaped leaf margin and has an excellent ability to absorb fine dust. *Platycladus orientalis*, an evergreen coniferous tree, has dense leaves from the bottom to the top of the tree and can block fine dust. These two types of trees were placed adjacent to the gymnasium. In the front area, *Platycladus orientalis*, evergreen, or deciduous shrubs were planted to form a layer. *Euonymus japonicus*, an evergreen broad-leaved shrub, has leaves with wax layers on both sides. *Viburnum erosum*, *Deutzia parviflora*, and *Weigela subsessilis*, deciduous broad-leaved shrubs, have hairy leaves. The ground surface was planted with ground cover plants and herbaceous flowers (*Gaura lindheimeri*, *Coreopsis drummondii*, *Liriope platyphylla*, *Iris sanguinea*, etc.) to prevent the respreading of subdued fine dust and to make the scenery beautiful. The construction of the fine dust reduction planting model was carried out after transplantation of the existing plantings and was completed on 27 May 2020.

**Table 3.** Information on plants at the experimental site after applying the fine dust reduction planting model.

| Division | Symbol | Scientific Name | Size [1] | No. |
|---|---|---|---|---|
| Evergreen trees | | *Platycladus orientalis* L. | H2.5 × W0.8 | 16 |
| Deciduous trees | | *Zelkova serrata* (Thunb.) Makino | H3.5 × R10 | 5 |
| Evergreen shrubs | | *Euonymus japonicus* Thunb. | H1.5 × W0.5 | 53 |
| Deciduous shrubs | | *Viburnum erosum* Thunb. | H1.0 × W0.4 | 24 |
| | | *Deutzia parviflora* Bunge | H1.0 × W0.3 | 42 |
| | | *Weigela subsessilis* (Nakai) L.H.Bailey | H1.0 × W0.4 | 16 |
| Groundcover plants and herbaceous flowers | | *Gaura lindheimeri* Engelm. and A.Gray | 8 cm | 142 |
| | | *Coreopsis drummondii* Torr. and A.Gray | 8 cm | 512 |
| | | *Liriope platyphylla* F.T.Wang and T.Tang | 8 cm | 43 |
| | | *Iris sanguinea* Donn ex Horn | 8 cm | 85 |
| | | *Pennisetum alopecuroides* (L.) Spreng. | 8 cm | 384 |
| | | *Miscanthus sinensis* 'Green Light' | 12 cm POT | 48 |
| | | *Miscanthus sinensis* 'Morning Light' | 12 cm POT | 48 |
| | | *Miscanthus sinensis* 'Variegatus' | 12 cm POT | 48 |
| | | *Miscanthus sinensis* 'Zebrinus' | 12 cm POT | 48 |
| | | *Hosta plantaginea* (Lam.) Asch. | 8 cm | 512 |
| | | *Dianthus chinensis* L. | 8 cm | 56 |

[1] H is tree height (m), W is crown width (m), R is root-collar caliper (cm), and POT is seedling pot.

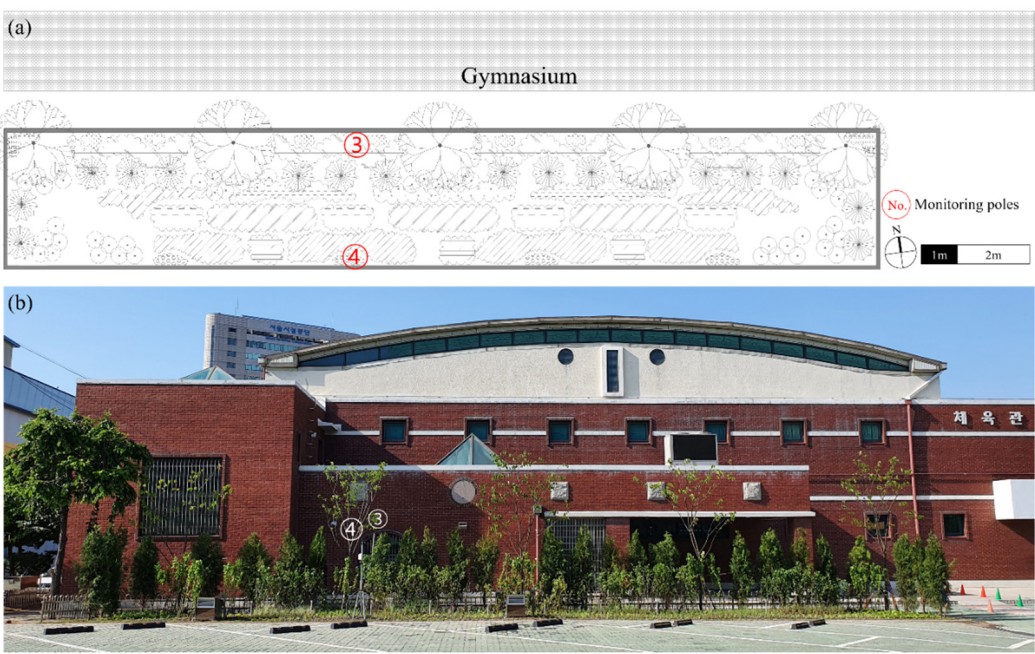

**Figure 4.** Diagram of the experimental site after applying the fine dust reduction planting model (**a**). Photo of the experimental site after applying the fine dust reduction planting model (**b**).

### 2.3. Monitoring Parameters and Devices

Environmental factors affecting the concentration of external fine dust include temperature, relative humidity, wind direction, and wind speed. Continental high pressure generally raises temperatures and stagnates the atmosphere to increase fine dust concentrations. When the temperature rises, the chemical reaction rate of ionic components such as sulfate, nitrate, and ammonium increases, resulting in more fine dust such as sulfur oxides ($SO_x$), nitrogen oxides ($NO_x$), and ammonia ($NH_3$) [29–32]. The relationship between $PM_{2.5}$ concentration and relative humidity is divided into three stages according to the relative humidity value. The concentration of $PM_{2.5}$ increases rapidly in areas with less than 45% relative humidity, gradually increases in areas with 45–70% relative humidity, and decreases in areas with more than 75% relative humidity [33]. $PM_{10}$ concentrations increase in the 0–45% relative humidity range but decrease as the relative humidity increases to more than 45% [33]. The wind speed and wind direction generated by the flow of air also have a great influence on the concentration of fine dust. Depending on the wind speed and wind direction, fine dust may be dispersed, or it may flow into a specific space and gather. Generally, the concentration of fine dust decreases as the wind speed increases, but when the ground wind speed exceeds 6 m/s, the concentration of fine dust increases due to respreading [34–36]. According to a study on the relationship between wind direction and fine dust in local space, the inflow of fine dust was observed to be high in the main direction of the wind [37]. Based on previous studies, $PM_{10}$, $PM_{2.5}$, temperature, relative humidity, wind speed, and wind direction were selected as external fine dust monitoring parameters.

The Smart Aircok Outdoor Type 1 measurement system designed by Aircok in Seoul 04799, Korea, which received the first grade from the Korea Testing and Research Institute and was designated by Korea as fine dust and weather factor measurement equipment, was used. This equipment measures $PM_{10}$, $PM_{2.5}$, temperature, and relative humidity in 1 min increments, and the collected data are stored in real time in the Amazon Web Service through LTE communication networks. According to the research results of the Ministry of Environment indicating that there is no difference in the concentration of fine dust according to the measurement height (2 m, 3.5 m, 5 m) [38], the monitoring equipment was installed at a height of 2 m on all monitoring poles (Figures 2–4 and Table 4). For wind monitoring, a wind speed and direction smart sensor designed by Onset Computer

Corporation in MA 02532, US was used. The average wind speed and average wind direction measured every minute with this device were stored on HOBO® data loggers designed by Onset Computer Corporation in MA 02532, US. Wind sensors and data loggers were installed on monitoring poles No. 1 and No. 4 (Figures 2–4, Table 4).

**Table 4.** Information on monitoring parameters and devices.

| Parameters | Unit | Measurement Range | Devices (Model Name) | Images |
|---|---|---|---|---|
| PM$_{10}$ | µg/m$^3$ | 0–1000 | Outdoor Air Quality Measuring Device (Smart Aircok Outdoor Type 1) | |
| PM$_{2.5}$ | µg/m$^3$ | 0–1000 | | |
| Temperature | °C | −10–60 | | |
| Relative humidity | % | 0–99 | | |
| Wind speed | mph | 0–170 | Davis® Wind Speed and Direction Smart Sensor (S-WCF-M003), HOBO® Micro Station Logger (H21-USB) | |
| Wind direction | ø | 0–355 | | |

### 2.4. Data Collection

The data collection period was divided into before and after the application of the fine dust reduction planting model at the experimental site. The data collection period before application was from 10 December 2019 to 13 May 2020. During this period, the control and experimental sites were not changed. The data collection period after applying the fine dust reduction planting model to the experimental group was from 27 May 2020 to 25 November 2020. Data from the control site during this period were not used beyond the scope of the study.

For statistical analysis of the 6 monitored parameters, data collected from monitoring equipment were processed. The PM$_{10}$, PM$_{2.5}$, temperature and relative humidity data collected every minute were converted into daily average values excluding values outside the measurement range or unmeasured values. The wind speed was converted from 1440 per minute wind speed values into daily average values. For wind direction, the observations collected at 24 time points were classified into 8 directions, and the wind direction with the highest number of observations per day was selected. When two or more wind directions were identified as the most frequent per day, the number of wind directions was added to the left and right wind directions, and the number of wind directions was selected. In this way, all monitoring parameters were converted into daily data.

### 2.5. Method for Verifying the Effectiveness of Fine Dust Reduction in Planting Zones

To verify the effect of fine dust reduction in the planting zone, a paired sample *t*-test was performed using IBM SPSS Statistics 25 for the PM$_{10}$ and PM$_{2.5}$ data from the control and experimental sites. The following hypotheses were established to verify the difference in the concentration of fine dust inside and outside the planting zone.

**Hypothesis 1 (H1).** *There is a significant difference in fine dust concentrations inside and outside the planting zone.*

To verify this hypothesis, the difference in the concentration of fine dust inside (No. 1) and outside (No. 2) of the control site was compared. For the experimental site, the data were divided before and after the application of the fine dust reduction planting model,

and the difference in the concentration of fine dust inside (No. 3) and outside (No. 4) of the planting zone was compared. The subhypotheses for H1 verification were as follows:

**Hypothesis 1-1 (H1-1).** *There is a significant difference in the concentration of fine dust inside and outside of the control site.*

**Hypothesis 1-2 (H1-2).** *Before the application of the fine dust reduction planting model, there is a significant difference in the concentration of fine dust inside and outside the experimental site.*

**Hypothesis 1-3 (H1-3).** *After the application of the fine dust reduction planting model, there is a significant difference in the concentration of fine dust inside and outside the experimental site.*

At the control site and the experimental before applying the fine dust reduction planting model, *Juniperus chinensis* 'Kaizuka' and *Prunus yedoensis* Matsum. were planted in a single structure. However, since the planting intervals and number of trees were different, the green coverage ratio of the tree canopy area to the site area was different. In the control site, the tree canopy area is about 20 m$^2$, and the site area is 36 m$^2$; therefore, the green coverage ratio is 55%. In the pre-experimental site, the tree canopy area was about 35.5 m$^2$, and the site area was 82.6 m$^2$, meaning the green coverage ratio was approximately 43%. To analyze the effect of reducing fine dust in planting areas with different green coverage ratios (the green coverage ratio in the control site is 55%, and the green coverage ratio in the pre-experimental site is 43%), the following hypotheses were set, and the value obtained by subtracting the internal fine dust concentration from the external fine dust concentration of the control site and the value obtained by subtracting the internal fine dust concentration from the external fine dust concentration of the experimental site were compared.

**Hypothesis 2 (H2).** *There is a significant difference in the amount of fine dust reduction depending on the green coverage ratio.*

Finally, the following hypotheses were set to verify the difference in fine dust concentration according to the planting structure.

**Hypothesis 3 (H3).** *There is a significant difference in fine dust concentration depending on the planting structure.*

To verify this hypothesis, the subhypotheses for H3 verification were as follows:

**Hypothesis 3-1 (H3-1).** *Inside the planting zone, there is a difference in fine dust concentration between the single structure and the multilayered structure.*

**Hypothesis 3-2 (H3-2).** *Outside the planting zone, there is a difference in fine dust concentration between the single-structure and the multilayered structure.*

**Hypothesis 3-3 (H3-3).** *There is a significant difference in the amount of fine dust reduction in the single-structure planting and the amount of fine dust reduction in multilayered structure planting.*

This study tried to verify the above hypotheses to secure the validity of a fine dust reduction planting model for a small urban green space.

### 3. Results and Discussion

*3.1. Observational Monitoring Data*

The fine dust forecasting grade has different standards for each country. Korea has strict standards for fine dust compared to other countries, such as the United States and

the United Kingdom [39]. Monitoring data in this study were interpreted based on Korea's criteria in Table 5.

**Table 5.** The fine dust forecasting grade of the United States, United Kingdom, and Korea.

| $PM_{2.5}$ $(\mu g/m^3)$ (Grade) | | | $PM_{10}$ $(\mu g/m^3)$ (Grade) | | |
|---|---|---|---|---|---|
| **US** | **UK** | **KOR** | **US** | **UK** | **KOR** |
| 0.0–12.0 (Good) | 0–35 (Low) | 0–15 (Good) | 0–54 (Good) | 0–50 (Low) | 0–30 (Good) |
| 12.1–35.4 (Moderate) | 36–53 (Moderate) | 16–35 (Moderate) | 55–154 (Moderate) | 51–75 (Moderate) | 31–80 (Moderate) |
| 35.5–55.4 (Unhealthy for sensitive groups) | 54–70 (High) | 36–75 (Unhealthy) | 155–254 (Unhealthy for sensitive groups) | 76–100 (High) | 81–150 (Unhealthy) |
| 55.5–150.4 (Unhealthy) | ≥71 (Very high) | 76–500 (Very unhealthy) | 255–354 (Unhealthy) | ≥101 (Very high) | 151–600 (Very unhealthy) |
| 150.5–250.4 (Very unhealthy) | - | - | 355–424 (Very Unhealthy) | - | - |
| 250.5–500.4 (Hazardous) | - | - | 425–604 (Hazardous) | - | - |

The observational data for the parameters monitored at the control site and the experimental site before applying the fine dust reduction planting model were the same, as shown in Figure 5. The average monthly concentration of $PM_{10}$ measured in the control and experimental sites was in the range of 25–40 µg/m³, which is moderate (Figure 5a). The $PM_{2.5}$ concentration was in the normal range (18–27 µg/m³) and showed a pattern such as $PM_{10}$ with each type of monitoring equipment (Figure 5b). The temperature was higher than the average temperature in Seoul (Figure 5c), and the relative humidity was generally lower than the relative humidity in Seoul (Figure 5d). This is because the study site and surrounding areas are classified as urban and dry areas on the landcover map. In the case of wind direction, south wind (S) and southeast wind (SE) were the dominant winds at the control site (Figure 5e), and southwest wind (SW) was the dominant wind at the experimental site (Figure 5f). Wind speed was measured frequently at the light air (0.3–1.5 m/s) and light breeze (1.6–3.3 m/s) levels at the control and experimental sites. For the relationship between the fine dust concentration and the wind direction, a $PM_{10}$ value in the unhealthy range was measured when the northwest wind (NW) or southeast wind (SE) was blowing at the control site (Figure 5g). The $PM_{2.5}$ of the control site showed unhealthy or very unhealthy ranges when northwest (NW), south (S), southeast (SE), and east (E) winds occurred. (Figure 5h). At the experimental site, $PM_{10}$ showed a good or moderate range regardless of the overall wind direction (Figure 5i), but in the case of $PM_{2.5}$, an unhealthy range was found when the wind was blowing overall, regardless of direction (Figure 5j). Both the control and experimental sites showed a high concentration of $PM_{2.5}$ when the south wind types (S, SW, SE) were recorded.

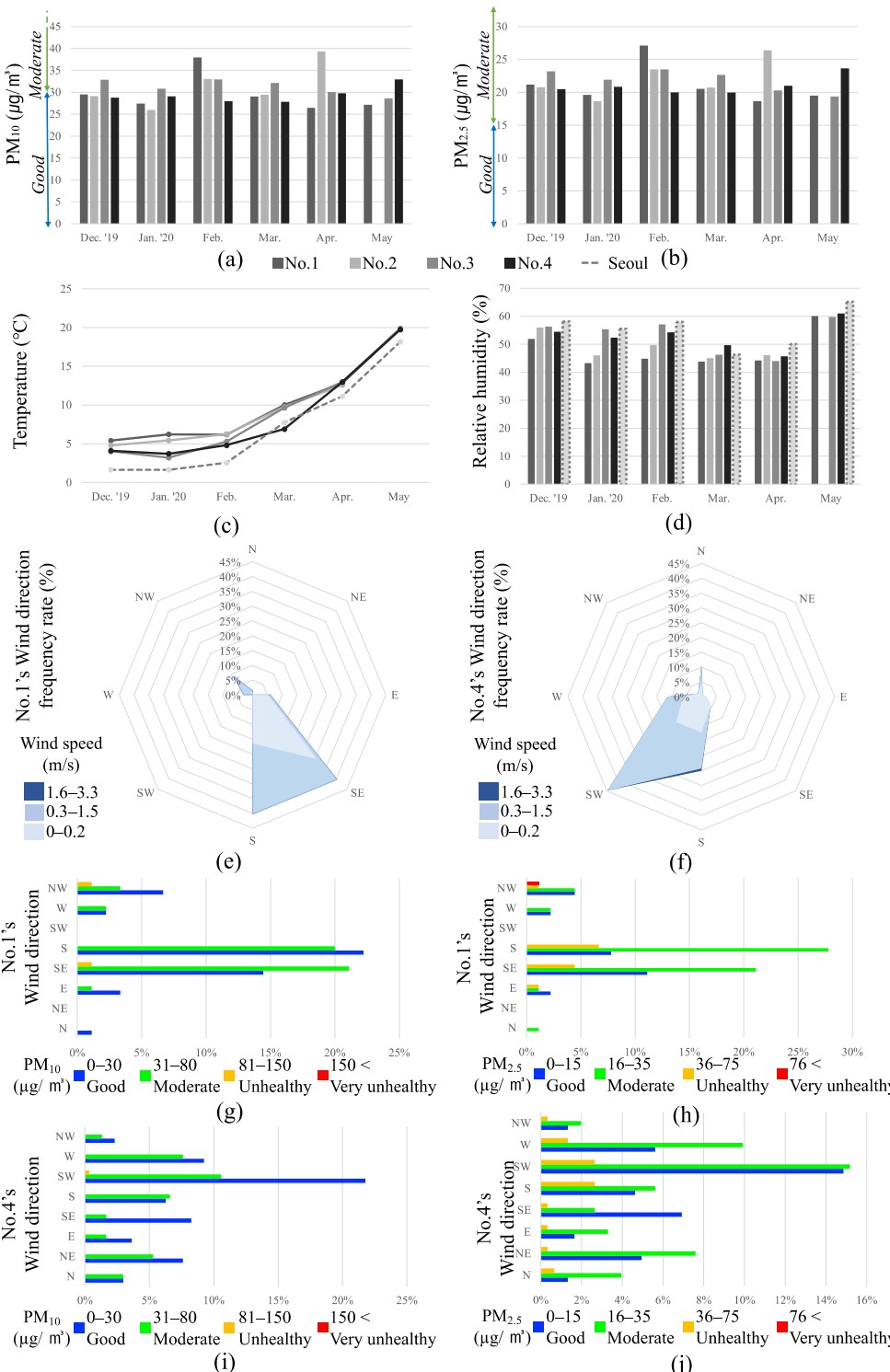

**Figure 5.** Monitoring data before applying the fine dust reduction planting model. No. 1 was the monitoring device inside the control site. No. 2 was monitoring device outside the control site. No. 3 was the monitoring device inside the experimental site. No. 4 was the monitoring device outside the experimental site. (**a**) is the average monthly PM$_{10}$. (**b**) is the average monthly PM$_{2.5}$. (**c**) is the average monthly temperature. (**d**) is the average monthly humidity. (**e**) is the wind rose at the control site. (**f**) is the wind rose at the experimental site. The direction of the longest spoke shows the wind direction with the greatest frequency. (**g**) is the average daily PM$_{10}$ ratio for the wind direction of the control site. (**h**) is the average daily PM$_{2.5}$ ratio for the wind direction of the control site. (**i**) is the average daily PM$_{10}$ ratio for the wind direction of the experimental site. (**j**) is the average daily PM$_{2.5}$ ratio for the wind direction of the experimental site.

The observational data for the monitored parameters at the experimental site after applying the fine dust reduction planting model are shown in Figure 6. The average monthly concentrations of $PM_{10}$ and $PM_{2.5}$ were lower from July to September, the rainy season in Korea, than in other months (Figure 6a,b). Monthly concentrations of $PM_{10}$ and $PM_{2.5}$ were lower in the planting zone (No. 3) than in the outside zone (No. 4). The difference in the concentration of fine dust between the inside and outside of the planting zone was more evident after the experiment than before the experiment. On the other hand, temperature and humidity were similar to the trends for Seoul (Figure 6c,d). The concentrations of $PM_{10}$ and $PM_{2.5}$ tended to decrease as the relative humidity in the atmosphere increased during the rainy season. The concentrations of fine dust of $PM_{10}$ and $PM_{2.5}$ outside the planting zone (No. 4) showed the lowest concentration in the section with a relative humidity of 75% or higher, and the concentration of fine dust showed a relative tendency to increase. In the inside of the planting zone (No. 3), although the relative humidity decreased in September after the rainy season, the concentrations of $PM_{10}$ and $PM_{2.5}$ were lower in September than in August. The concentration of fine dust in the planting zone (No. 3) was less affected by relative humidity than the concentration of fine dust outside the zone (No. 4). The wind direction at the experimental site after the experiment was different from that before the experiment. While southwest wind (SW) dominated before the application of the fine dust reduction planting model, various wind directions, such as southwest (SW), west (W), northeast (NE), and southeast (SE) winds, were observed after the experiment (Figure 6e). When wind encounters an obstacle, the air flow in the front, back, and side of the obstacle changes [40,41], and the multilayered fine dust reduction planting model can be seen as an obstacle. $PM_{10}$ was measured in at nearly good or moderate ranges regardless of the overall wind direction (Figure 6f), but in the case of $PM_{2.5}$, an unhealthy range was found when almost all wind directions were observed (Figure 6g). Compared with the graphs of the experimental group before the experiment (Figure 5i,j), the ratio of fine dust in the good or moderate range flowing in from various wind directions increased after applying the fine dust reduction planting model (Figure 6f,g). In particular, $PM_{2.5}$ decreased by approximately 1% in the unhealthy range when south and north winds were observed. From these results, it can be determined that the fine dust reduction planting model plays a role in effectively blocking, absorbing, and adsorbing fine dust.

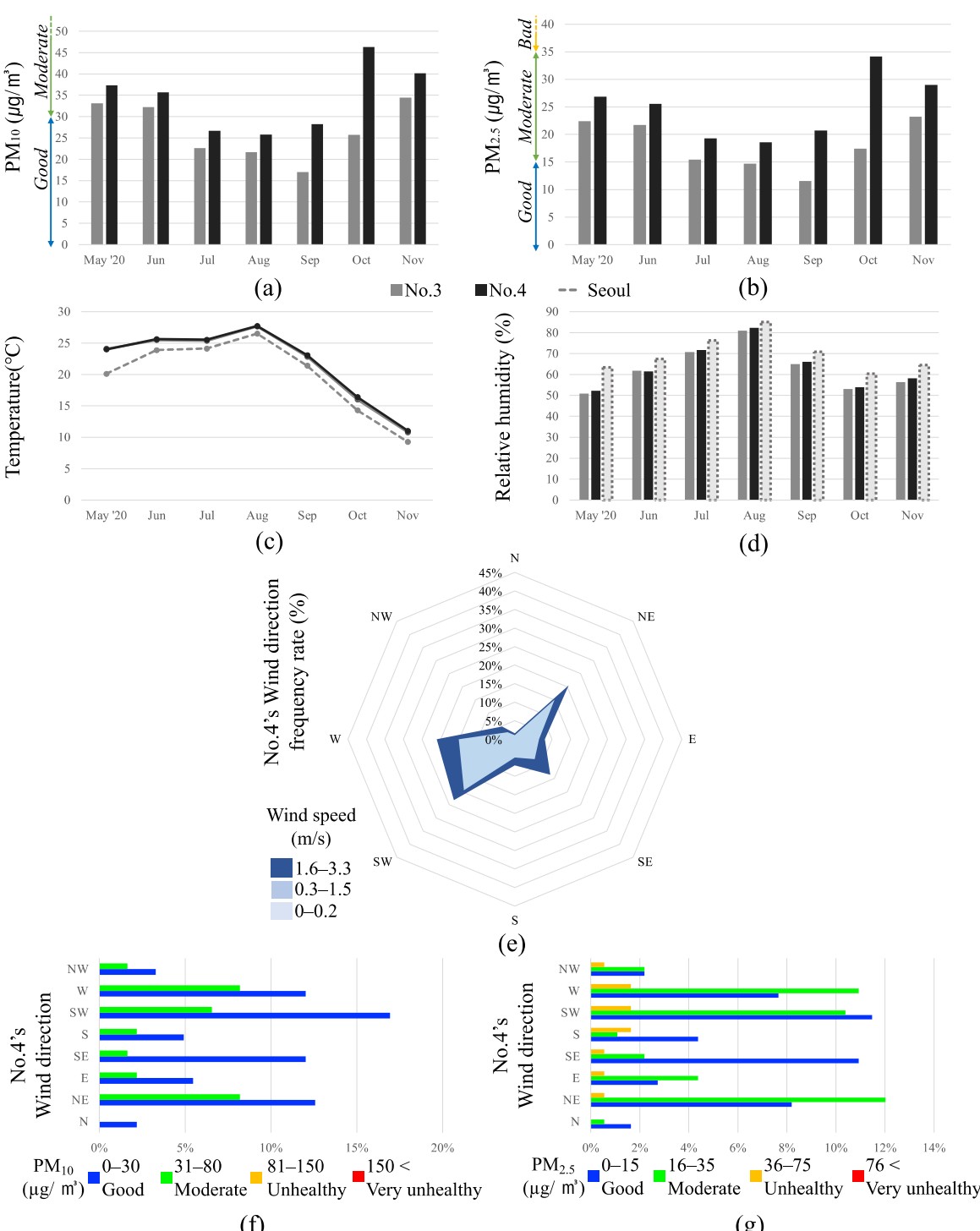

**Figure 6.** Monitoring data after applying the fine dust reduction planting model (27 May 2020–25 November 2020). No. 3 was the monitoring device inside the experimental site. No. 4 was the monitoring device outside the experimental site. (**a**) is the average monthly PM$_{10}$. (**b**) is the average monthly PM$_{2.5}$. (**c**) is the average monthly temperature. (**d**) is the average monthly humidity. (**e**) is the wind rose. The direction of the longest spoke shows the wind direction with the greatest frequency. (**f**) is the average daily PM$_{10}$ ratio for wind direction. (**g**) is the average daily PM$_{2.5}$ ratio for the wind direction.

### 3.2. Verification of Differences in Fine Dust Concentrations Outside and Inside Planting Zones

The results of the paired sample *t*-test for the difference in fine dust concentration outside and inside the planting zone are the same as those in Table 6. The null hypothesis was rejected for all three subhypotheses of H1, indicating that there was a significant difference

in the concentration of fine dust outside and inside the planting zone. In the case of the control site (H1-1) and the experimental site after the experiment (H1-3), the concentration of fine dust outside the planting zone was higher than that inside. Consistent with the results of this study, several previous studies have demonstrated that the concentration of fine dust inside is lower than that outside of the planting zone [42,43]. However, in the case of the experimental site before the experiment (H1-2), the concentration of fine dust inside the planting zone was higher than that outside. While the planting zone of the control site and the experimental site after the experiment had a positive effect on the reduction of fine dust, what was the reason for the negative effect on the reduction of the fine dust in the planting zone of the experimental site before the experiment? The control site and the pre-experimental site had the same species, but the planting intervals were different. The green coverage ratio of the control site is higher. The species and planting structure are different for the pre- and post-experimental sites. After the experiment, the species of the experimental site were more diverse, dense, and multilayered. The following analysis was conducted to verify whether the difference in the density and structure of the planting can determine the difference in the concentration of fine dust inside and outside the planting zone.

**Table 6.** The results of paired sample *t*-tests of the difference of fine dust concentration between outside and inside the planting zone.

| Hypothesis No. | Subject | N | Mean ($\mu g/m^3$) (SD) | | Difference ($\mu g/m^3$) | *t*-Value | *p*-Value [1] |
|---|---|---|---|---|---|---|---|
| | | | Outside | Inside | | | |
| H1-1 | C_ $PM_{10}$ | 88 | 32.83 (17.93) | 30.55 (18.12) | 2.28 | 3.11 | 0.003 |
| | C_ $PM_{2.5}$ | 88 | 22.97 (12.57) | 21.72 (13.03) | 1.24 | 2.63 | 0.010 |
| H1-2 | E_Pre_ $PM_{10}$ | 132 | 29.78 (14.86) | 32.34 (18.28) | −2.55 | −5.27 | 0.000 |
| | E_Pre_ $PM_{2.5}$ | 132 | 21.27 (10.78) | 22.71 (13.27) | −1.44 | −3.83 | 0.000 |
| H1-3 | E_Post_ $PM_{10}$ | 179 | 33.67 (24.15) | 25.29 (14.62) | 8.38 | 5.89 | 0.000 |
| | E_Post_$PM_{2.5}$ | 179 | 24.44 (18.04) | 17.12 (9.68) | 7.32 | 6.70 | 0.000 |

[1] $p < 0.05$. C is the control site. E is the experimental site. Pre means before applying planting model for reducing fine dust. Post means after applying the planting model for reducing fine dust.

*3.3. Verification of Differences in the Amount of Fine Dust Reduction According to the Green Coverage Ratio*

The results of the analysis showed that the control site with a higher green coverage ratio (55%) was effective in reducing fine dust compared to the pre-experimental site (43%) (Table 7). The green coverage ratio affects the number of anions generated in the plant and is involved in the photosynthesis and transpiration of the plant [44,45]. The physiological actions of plants affect the hair, gloss, wax layer, resin, and flexibility of the leaf surface, which are the main factors in the reduction of fine dust [27]. In other words, a high green coverage ratio leads to an increase in leaves and stems, which are effective in absorbing fine dust [46,47]. This study found that a green coverage ratio of more than 50% was effective in reducing fine dust.

**Table 7.** The results of the paired sample *t*-test of the difference in fine dust reduction between the control site and the pre-experimental site.

| Hypothesis No. | Subject | N | Mean (μg/m³) (SD) | | Difference (μg/m³) | *t*-Value | *p*-Value [1] |
|---|---|---|---|---|---|---|---|
| | | | Control Site | Pre-Experimental Site | | | |
| H2 | Pre_PM$_{10}$ reduction | 73 | 2.21 (6.78) | −3.01 (4.91) | 5.23 | 6.13 | 0.000 |
| | Pre_PM$_{2.5}$ reduction | 73 | 1.16 (4.34) | −1.80 (3.65) | 2.96 | 5.24 | 0.000 |

[1] $p < 0.05$. Pre means before applying planting model for reducing fine dust.

### 3.4. Verification of Differences in Fine Dust Concentration and Reduction Amount According to Planting Structure

To verify the effectiveness of the planting model for reducing fine dust, the fine dust concentrations before and after the experiment at the experimental site were compared. As a result of the analysis, inside the experimental site, the concentration of fine dust before the experiment was measured to be higher than that after the experiment (Table 7, H3-1). However, there was no significant difference in the concentration of fine dust outside the experimental site before and after the experiment (Table 7, H3-2). On the other hand, it was found that the amount of fine dust introduced from outside the experimental site to be reduced inside the experimental site was greater after the experiment than before the experiment (Table 8, H3-3). This study found that fine dust is reduced inside the planting zone when various types of plants are planted in multilayered structures. Groundcover plants, herbaceous flowers, and shrubs that form the underlying structure settle the fine dust introduced into the vegetation and prevent respreading of fine dust. In particular, shrubs with dense branches and leaves also acted as filters to adsorb and absorb fine dust. The trees that form the upper structure played a role in adsorbing and absorbing fine dust of 1.5 m or more from the ground. The multilayered planting zone not only reduces the concentration of fine dust by the function of plants but also creates the effect of spreading fine dust to the surroundings by changing the direction of the wind in various ways (Figure 6e). As the plant species and planting model proposed in this study proved to have a statistically significant effect on the reduction of fine dust, the feasibility of applying this method to spaces requiring fine dust reduction in the future was verified.

**Table 8.** The results of the paired sample *t*-test of the difference in fine dust concentration and reduction amount before and after applying the fine dust reduction planting model at the experimental site.

| Hypothesis No. | Subject | N | Mean (μg/m³) (SD) | | Difference (μg/m³) | *t*-Value | *p*-Value [1] |
|---|---|---|---|---|---|---|---|
| | | | Pre | Post | | | |
| H3-1 | In_PM$_{10}$ | 146 | 32.31 (17.82) | 24.39 (13.12) | 7.91 | 4.26 | 0.000 |
| | In_PM$_{2.5}$ | 146 | 22.67 (12.91) | 16.52 (8.69) | 6.15 | 4.71 | 0.000 |
| H3-2 | Out_PM$_{10}$ | 136 | 29.71 (15.08) | 31.06 (20.73) | −1.34 | −0.62 | 0.531 |
| | Out_PM$_{2.5}$ | 136 | 21.23 (10.93) | 22.49 (15.43) | −1.25 | −0.79 | 0.426 |
| H3-3 | PM$_{10}$ reduction | 128 | −2.51 (5.63) | 7.21 (15.49) | −9.72 | −6.96 | 0.000 |
| | PM$_{2.5}$ reduction | 128 | −1.40 (4.36) | 6.37 (11.98) | −7.78 | −7.33 | 0.000 |

[1] $p < 0.05$. In means inside the planting zone of the experimental site. Out means outside the planting zone of the experimental site. Pre means before applying the fine dust reduction planting model. Post means after applying the fine dust reduction planting model.

## 4. Conclusions

Recently, as urban green infrastructure has been shown to be effective in reducing fine dust, the demand to create green infrastructure to cope with the fine dust problem for children's health is gradually increasing. In this study, a fine dust reduction planting model was developed in an elementary school forest in a socially vulnerable area. This study may lead to important implications an to be considered when creating a small urban green space to reduce fine dust in socially vulnerable areas. First, it is necessary to sufficiently analyze various factors that affect the generation of fine dust, such as the arrangement and height of buildings around the target site and the weather conditions. The smaller the green space is, the greater the change in the concentration of fine dust responds to the surrounding environment. Therefore, various analyses, for example, of the area and width of the green zone, the status of existing plantings, the distance from buildings, the creation of shadows due to buildings, the land cover status, people's movement, weather conditions (temperature, humidity, wind speed, and wind direction), and the occurrence of fine dust, are needed [48]. In this study, a fine dust reduction planting model was proposed based on the existing status analysis (Figure 4) of the experimental site for a sufficient period, and the effect was verified.

Second, multilayered planting using various plant types should be planned. Plants have different functions, such as fine dust adsorption, absorption, and blocking, and preventing dust from respreading according to plant type, including evergreen or deciduous trees, evergreen or deciduous shrubs, groundcover plants, and herbaceous flowers. Evergreen trees have dense leaves to block the inflow of fine dust on the ground, and deciduous trees form a wide canopy to block the descent of floating fine dust from a height of 3 m to 5 m. Evergreen or deciduous shrubs use sticky or hairy leaves and dense branches to absorb fine dust from the outside. Groundcover plants and herbaceous flowers prevent respreading of fine dust that has sunk to the ground. If a single type of plant is planted, the effect of fine dust reduction can be reduced. In fact, this study found that the experimental site reduced more fine dust after the experiment where all types of plants were planted in multilayered plant structures than the control and pre-experimental sites planted in a single plant structure of evergreen or deciduous trees (Table 6). Particularly, Table 8 (H3-3) revealed that the multilayered planting could reduce $PM_{10}$ by 9.72 $\mu g/m^3$ and $PM_{2.5}$ by 7.78 $\mu g/m^3$ more than the single-layered planting. Therefore, the plant type for reducing fine dust should be selected considering the current state of fine dust generation in the target area, and multilayered planting should be planned as much as possible.

Third, the multilayered fine dust reduction planting model can change the surrounding microclimate, such as the humidity and wind direction. In this study, after applying the fine dust reduction planting model, the increase in $PM_{2.5}$ concentration was reduced in the area with a relative humidity of 50–62% compared to that in the 45–70% section reported in the previous study (Figure 5). This means that the area with a relative humidity contributing to the increase in ultrafine dust concentration was reduced by the multilayered planting. In addition, the multilayered planting dispersed the wind direction to reduce the concentration of fine dust flowing into the planting zone. These results suggest that fine dust can be reduced by changing the microclimate by multilayered planting.

The fine dust reduction planting model proposed in this study represents the first exploration of the optimal species and planting structure for fine dust reduction in small urban green spaces. It was also found that small-scale planting zones can reduce fine dust. The results and discussion of this study are expected to help establish strategies to effectively respond to the issue of fine dust in socially vulnerable areas where there is a relative lack of open green space.

Meanwhile, the limitations of this study are as follows: Most plants proposed as multilayered plants are easy to supply and demand in Korea. Therefore, it may be difficult to supply the plants in other countries because the climate and landscape markets are different from those of Korea. Nevertheless, the emphasis in this study is that multilayered structures in which various varieties are harmonized are more efficient in reducing fine

dust than planting consisting only of tall trees. Therefore, depending on the type of plant discussed in this study, it would be necessary to select the right species for each local environment. In a follow-up study, research is needed on the effect of reducing fine dust by individual tree species type (deciduous or conifers trees) and the effect of reducing fine dust according to various planting patterns, intervals, and layered structures, which were not covered in depth in this study. In addition, it is necessary to verify the effectiveness of the planting model in areas where much fine dust is generated due to adjacent roads or industrial facilities. If follow-up studies related to this are carried out in the long term, it is expected that more effective and concrete plans to reduce fine dust in cities can be established.

**Author Contributions:** Conceptualization, Y.C.; methodology, Y.C. and E.J.; validation, J.C.; formal analysis, Y.C. and E.J.; investigation, Y.C. and E.J.; resources, Y.C. and E.J.; data curation, Y.C. and E.J.; writing—original draft preparation, Y.C.; writing—review and editing, J.C.; visualization, Y.C. and E.J.; supervision, J.C.; project administration, J.C.; funding acquisition, J.C. All authors have read and agreed to the published version of the manuscript.

**Funding:** This study was carried out with the support of the R&D Program for Forest Science Technology (Project No. 2019153B10-2121-0101) provided by the Korea Forest Service (Korea Forestry Promotion Institute). This study was supported by the National Research Foundation of Korea (NRF) grant funded by the Korea government (MEST) (No. 2020R1C1C1007165) and supported by an OJERI (OJEong Resilience Institute) Grant.

**Institutional Review Board Statement:** Not applicable.

**Informed Consent Statement:** Not applicable.

**Data Availability Statement:** Data sharing not applicable.

**Conflicts of Interest:** The authors declare no conflict of interest.

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
