# Peer review of "Development and Verification of the Effectiveness of a Fine Dust Reduction Planting Model for Socially Vulnerable Area"

_sustainability, doi:10.3390/su13168820_

Round 1
Reviewer 1 Report
Please check the attached file.

Author Response
Thank you very much for your comments and insights that will help improve our paper. We found your suggestions and comments very beneficial and insightful. We reflected most of your suggestions and comments with the amendments summarized of an attachment file.

Reviewer 2 Report
Dear Authors,
Thank you very much for submitting your very interesting work to 'Sustainability'! I enjoyed reviewing your paper as it reminded me of the deposition studies carried out for industrial dust coming out of cement factories. However, I think that the concept of study is overall good but you might also investigate (further analysis based on already collected data), not only planting structure but also the species types - deciduous and coniferous. Some years ago the cement factories and heavy industries overall imposed a huge problem (as well as deposition from heavy industries which led to acid rains and trees dieback) which was solved by planting conifers in their vicinity to improve air quality. Of course, this was just one of the solutions for West Germany and the USSR at that time because the filters were introduced which captured the dust. During these experimental studies it was found that conifers capture more dust than deciduous species which could be interesting to compare with regard to finer dust particles in an urban environment. Having the exercise done by deciduous and conifers might confirm previous findings or provide new insight to this particular dust type.
Thank you in advance for having my comments and suggestions into consideration!
Kind regards,
Reviewer
Author Response
Dear Reviewer,
We appreciate your kind words and comments. Many studies have shown that conifers with dense leaves are effective in improving air quality by adsorbing many fine dust particles. We agree that research is needed to analyze the effect of reducing fine dust according to the species types of trees (deciduous and conifers) not only in industrial areas, but also in urban living areas. So we added the aforementioned to the conclusion (L 476-479). Once again, we would like to thank you for recommending a topic that may be helpful for the follow-up of the study.
Reviewer 3 Report
The observation period before the experiment and the experiment were not comparable due to the weather conditions and plant phenology in the vicinity of the experiment area, which is confirmed by the results of the analyzes. For these two periods, the t-test analysis for paired data is correct.
Comparison of the control area with the experimental area should not be made by a paired t-test. These are independent areas with different spatial location. They are exposed to other influences of the wind direction (Fig. 1a).
Whether the LAI (leaf area index) was measured, the surface coverage index is not appropriate for tall plants. The authors emphasize that it is a multi-layer structure and therefore it should be described with an appropriate indicator.
Table 5, 6, 7 - supplement the descriptions with the sample size and the value of the standard deviation
Author Response
We greatly appreciate your comments and suggestions. We found your suggestions and comments very helpful and insightful, the paper is now noticeably better. We considered your comments carefully and summarized the amendments of an attachment file.

Reviewer 4 Report
With the changing global climate and global warming becoming more and more impactful, this article comes at the right time to provide suitable solutions to our dusty vulnerable cities which constitute real heat islands. This forces governments to dust reduction policies including regreening. The re-greening of the countryside as well as that of the cities is increasingly seen as the best way to fight against global warming, variability and change.
Very present in the air of industrialized countries, particularly in urban areas, fine particles have an impact on our health. Who are the main emitters? And what are the dangers of breathing these particles? In cities, the most vulnerable environments are those exposed to occurrences of heat, dust and fine particles PM10 and PM2.5. These are the schools, because children are the first victims, dense roads, etc.
Also, I found that during past years, “Korea figured among the OECD countries with the highest share of population exposed to excessive PM2.5 (atmospheric particulate matter that have a diameter of less than 2.5 micrometers) concentrations and PM2.5 concentration level in Seoul is about two times higher than the WHO’s guidelines or the levels of other major cities in developed countries. A number of countermeasures have been recently introduced to address such challenges, including a tightening of air quality standards and increasing local inspection and enforcement capacity”.
Such a finding could explain the opportunity raised up by this study. We know that plants are real screens against heat, dust and winds. Therefore their use in a reduction of fine particles is timely and desired. This Korean study demonstrates the effective application for fine particle reduction, a reduction that may just as support the fight against COVID-19, as this study can be applied at all scales Worldwide. The authors in their experiment gave the types of plants to apply contextually (table 3). The results show the effectiveness of the proposed model in reducing the absorption and concentration of dust and there may be very few in the world. In my opinion, such a study is not useful only for Korea, because the African cities of the Sahel and the Third World as well as the countryside can benefit from the proposed experience and this is what makes this article more valuable. The reviewer would like to thank the authors for the quality of writing and experimentation while raising two minor concerns;
The conclusion seems long and it should deliver ‘essential results with some statistics from Tables 6 and 7. The limits of the study were not specified by the authors.
Author Response
Dear Reviewer,
We greatly appreciate your comments and kind suggestions. We provided and improved the paper based on the recommendations. We supplemented the discussion by mentioning statistical results (L49-451, 455-458) and added limitations to the study and follow-up study (L469-479), instead of reducing duplicate content in the conclusion.
Round 2
Reviewer 1 Report
The reviewer appreciates the authors' efforts to improve the quality of the manuscript. The manuscript, as a whole, is now more scientifically sound and findings are described logically.